# Clinicohematological and molecular analysis of hemoglobin D syndrome and unknown variants in the hemoglobinopathy spectrum of Sindh, Pakistan

Sunila Tashfeen[1]*, Ikram Din Ujjan[1], Hina Shaikh[2], Muhammad Arif Sadiq[3], Feriha Fatima Khidri[4], Ali Raza Rajput[2], Ali Muhammad Waryah[2]

1 Department of Pathology, Liaquat University of Medical and Health Sciences, Jamshoro, Pakistan, 2 Department of Molecular Biology and Genetics, Liaquat University of Medical and Health Sciences, Jamshoro, Pakistan, 3 Pak Emirates Military Hospital, Rawalpindi, Pakistan, 4 Department of Biochemistry, Bilawal Medical College, Liaquat University of Medical and Health Sciences, Jamshoro, Pakistan.

* sunila.tashfeen@yahoo.com

## Abstract

### Objectives:

Hemoglobinopathies are prevalent monogenic disorders resulting from genetic abnormalities in globin genes, significantly impacting health. β-thalassemia is particularly common in Pakistan, but data on other hemoglobin variants remain limited. This study aimed to investigate HbD syndrome, identify unknown variants, and examine the clinicohematological and molecular profiles of hemoglobinopathies in Sindh, Pakistan.

### Methods:

A prospective cross-sectional study was conducted from January 2021 to January 2023 at Liaquat University of Medical and Health Sciences (LUMHS), Jamshoro, Pakistan. Blood samples were collected from across Sindh, Pakistan and analyzed for hemoglobinopathies using hematological tests (CBC, peripheral blood smear), cation exchange high-performance liquid chromatography (CE-HPLC) and molecular analysis to confirm HbD and identify rare variants. Data were analyzed using SPSS v. 27.

### Results:

Out of 4783 chromatograms analyzed, 1563 (32.7%) were diagnosed with hemoglobinopathies. The most common conditions included β-thalassemia (81.4%), hemoglobin (Hb) variants (11.2%), and hereditary persistence of fetal hemoglobin (7.4%). HbD was found in 2.1% of cases, with HbD syndromes being the most prevalent among Hb variants (56.6%). Sickle cell disorders followed with a frequency of 32%,

**Data availability statement:** All relevant data are within the manuscript and its supporting information file.

**Funding:** The author(s) received no specific funding for this work.;

**Competing interests:** The authors have declared that no competing interests exist.

and HbQ, HbE, and HbC were less common. Molecular analysis confirmed the HbD Punjab variant and identified an additional four mutations, i.e., one rare β-thalassemia mutation and three Hb variants including Hb Hinsdale, Hb Renert and Hb Takasago.

## Conclusion:

Hb D Punjab is the most prevalent hemoglobin variant in Sindh, Pakistan, followed by HbS and HbQ. Molecular analysis is essential for accurate diagnosis and identifying rare variants. Integrating HbD detection into screening programmes and genetic counselling can help prevent hemoglobinopathies. (S1 Abstract Graphic).

## Introduction

Hemoglobinopathies represent the most prevalent monogenic disorders, with an approximate carrier rate of 7% among the world population, contributing significantly to childhood morbidity and mortality. According to the World Health Organization (WHO), every year, 300,000–400,000 infants are born with severe inherited disorders of hemoglobin (Hb) [1]. Genetic diseases linked to Hb synthesis can result from DNA variants in or near the globin genes, with over 95% of cases resulting from genetic abnormalities of the β-globin gene (HBB). These disorders may either lead to a reduction in the production of globin chains (alpha or beta), known as thalassemia syndromes, or the generation of abnormal Hb, referred to as Hb variants [2,3]. The main structural Hb variants include HbS, HbD, HbE, and HbC whereas Hb Lepore, HbJ Meerut, HbH disease and HbQ India have been reported less frequently in various populations. Certain other forms exhibits characteristics from both categories, such as β0/β + -thalassemias, HbSC disease, and HbE α-thalassemias [4]. According to HbVar database, there are currently 1870 human Hb variants and thalassemia mutations identified [5]. The prevalence and types of Hb variants vary significantly based on geographic location and ethnicity, and the diagnosis of these variants is crucial for investigating hemoglobinopathies and preventing birth defects [6].

Among Hb disorders, β-thalassemia emerges as the predominant single-gene disorder in Pakistan [7], with a prevalence of 5–8% for the β-thalassemia trait, leading to 5000–9000 annual births with β-thalassemia major. Concurrently, approximately 50,000 patients are undergoing treatment nationwide [2]. Although carriers of Hb disorders typically show no clinical signs or symptoms, those in the homozygous or compound heterozygous state may develop symptoms leading to life-threatening consequences, necessitating lifelong dependence on blood transfusions. Though hematopoietic stem cell transplantation remains a curative treatment, its high cost makes it unaffordable for many patients in low- and middle-income countries [8,9]. While substantial attention has been given to β-thalassemia, limited data exist on other hemoglobin (Hb) variants, such as HbD and rare structural variants like HbQ India, particularly in the geographically and ethnically diverse population of Sindh, Pakistan.

HbD syndromes may manifest as heterozygous HbD trait, HbD thalassemia, HbSD disease, and homozygous HbD disease. Heterozygous HbD is generally a benign condition without apparent illness; however, if associated with HbS or β-thalassemia, it may result in sickling disease and moderate hemolytic anemia [10,11]. HbQ India (HbA1:c.193G > C), is an alpha-chain structural Hb variant caused by an amino acid substitution of histidine for aspartic acid at codon 64 of the alpha1-globin gene. It can present as heterozygous, homozygous or co-inherited with β-thalassaemia [4]. Despite advancements in molecular techniques, Pakistan lacks comprehensive data on the prevalence, molecular characterization, and clinicohematological profiles of hemoglobin variants, creating a significant gap in understanding the genetic burden of hemoglobinopathies. This gap hinders progress in genetic counseling, prevention, and personalized treatment strategies.

Given the limited data on hemoglobinopathies and Hb variants, this study aimed to investigate HbD syndrome, unknown variants, their assessment at the molecular level, and the clinicohematological profile associated with hemoglobinopathies in Sindh, Pakistan. The findings are expected to enhance diagnostic accuracy, guide effective management strategies, and contribute to national efforts in combating the burden of hemoglobinopathies.

## Methods

### Study design and setting

This prospective cross-sectional study was conducted at the Department of Pathology, Liaquat University of Medical and Health Sciences (LUMHS), Jamshoro, Pakistan, from January 2021 to January 2023 following approval from the Research Ethics Committee (LUMHS/REC/766; dated 19-02-2019). The Diagnostic and Research Laboratory, LUMHS, serves as a referral laboratory, receiving samples from five collection points within Hyderabad city and 26 collection centres across all districts of Sindh province (Fig 1). Thus, the collected samples represent the broader population of Sindh, Pakistan. Written informed consent was obtained from patients or their next of kin following a thorough explanation of the study protocol. Patient details, history, and clinical examination findings were recorded in a proforma.

### Inclusion and exclusion criteria

The inclusion criteria comprised patients residing in the Sindh, Pakistan, who were recommended for Hb studies for the investigation of anemia, hypochromic microcytic blood picture, and hemoglobinopathies, irrespective of age and gender. Recent blood transfusion within the last 3 months, known cases of thalassemia and repeat testing were considered exclusion criterion.

### Hematological analysis

For hematological investigations, 3 ml of blood sample was collected in purple top anticoagulant tubes containing ethylene diamine tetraacetate (EDTA). Complete blood counts (CBC), including red cell indices such as red blood cell (RBC) count, Hb, haematocrit (HCT), mean corpuscular volume (MCV), mean corpuscular hemoglobin (MCH) and mean corpuscular hemoglobin concentration (MCHC), were analyzed using the Sysmex XN-1000i. Peripheral blood smears were stained with Leishman's stain, examined for RBC morphology, and graded according to standard criteria. For Hb identification and quantification, cation exchange high-performance liquid chromatography (CE-HPLC) [12] was performed using the Bio-Rad Variant II (Bio-Rad Laboratories Inc., Hercules, CA, USA). Peaks observed in the chromatograms were identified by referencing retention time (RT), relative percentage, and area under the curve. The retention times of HbA, HbA2, HbF, HbD, HbS, and the unknown window were compared to known retention times to quantify both normal and variant Hb. An RT of 3.90 to 4.30 min was used to identify HbD. All samples with peaks in the unknown window were further investigated.

Iron deficiency was determined using serum ferritin levels (<20 ng/ml). The cut-offs for anemia were as follows: adult males (mild: Hb 11.0-12.9 g/dl; moderate: 8.0-10.9 g/dl; severe: < 8.0 g/dl), adult females (mild: Hb 11.0–11.9 g/dl;

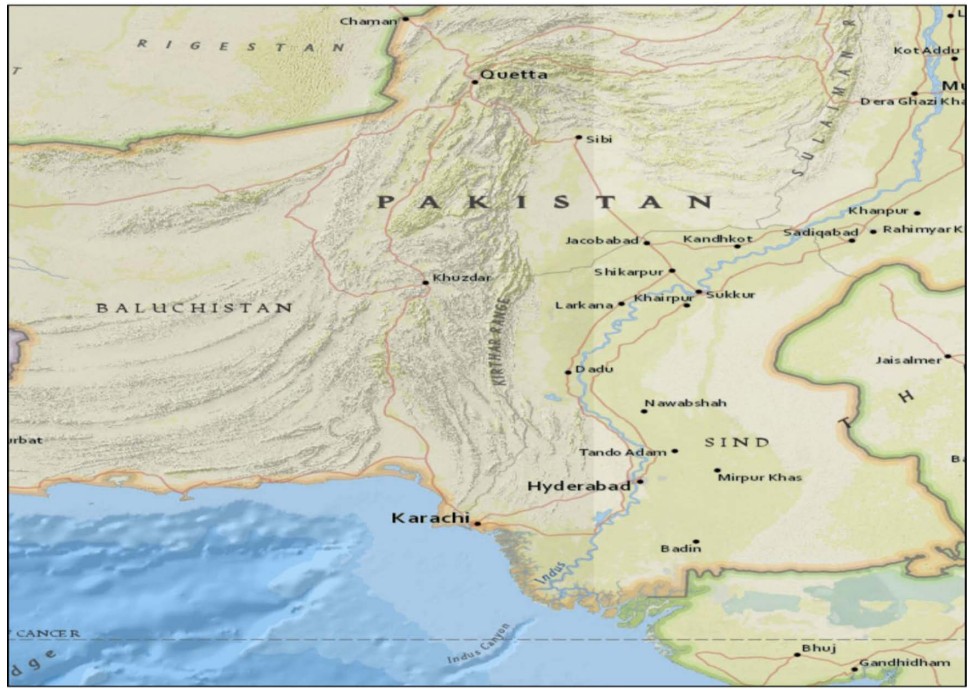

**Fig 1. Map of Sindh Province, showing the distribution of samples received by the 31 collection centers of the Diagnostic and Research Laboratory, Liaquat University of Medical and Health Sciences (LUMHS), Jamshoro, Pakistan, covering 16 districts within Sindh Province for hemoglobin studies.**

moderate: 8.0–10.9 g/dl; severe: Hb < 8.0 g/dl), and children according to age group, as defined by World Health Organization [13]. For cases with elevated MCV levels observed on peripheral blood films, additional testing was conducted, including measurements of serum vitamin B12 and folate levels.

## Molecular analysis

For molecular analysis, samples from patients with HbD and unknown variants were examined. Genomic DNA was extracted from 3 ml of blood collected in EDTA tubes using an inorganic method [14] and purity were assessed via spectrophotometry. For HbD analysis, exons of the HBB gene were amplified using pairs of forward (F) and reverse (R) primers with following sequences:

HBB1-F: 5' CGG CTG TCA TCA CTT AGA CC 3'
HBB1-R: 3' CCT AAG GGT GGG AAA ATA GACC 5'.
HBB2-F: 5' CGT GGA TGA AGT TGG TGG TG 3'
HBB2-R: 3' ACC CTG TTA CTT ATC CCC TTCC 5'.
HBB3-F: 5' ACA ATC CAG CTA CCA TTC TGC 3'.
HBB3-R: 3' CTG ACC TCC CAC ATT CCC TT 5'.

Molecular analysis of the HbD gene was conducted using capillary sequencing on the Applied Biosystem ABI PRISM 3130 Genetic Analyzer (Foster City, California, USA). Chromatogram sequences of the HBB gene exons 1,2 and 3 were aligned using the BLAST tool on online software blast.ncbi.nlm.nih.gov.

Unknown variants, suspected to be HbQ India, based on RT of 4.68 min were further examined using the amplification refractory mutation system (ARMS) PCR [15] targeting the α1 globin gene, as published previously [16].

## Statistical analysis

Data analysis was performed using SPSS v. 27. Continuous variables were expressed as mean ± standard deviation and range, while categorical variables were presented as frequencies and percentages. The distribution of hemoglobinopathies by age and gender was statistically analyzed using the chi-square test or Fisher's exact test, where applicable. A p-value < 0.05 was considered significant.

## Results

### Distribution of haemoglobinopathies

Among the 4,783 chromatograms analyzed from study participants across various districts of Sindh, 1,563 (32.70%) were diagnosed with haemoglobinopathies, while 3,220 (67.30%) exhibited normal haemoglobin chromatograms. The distribution of hemoglobinopathies by age and gender, along with normal chromatograms, is presented in Table 1.

The identified haemoglobinopathies included β-thalassaemia in 1,272 (81.38%), Hb variants in 175 (11.19%), and hereditary persistence of fetal haemoglobin (HPFH) in 116 (7.4%) of the patients. The overall frequency of HbD was 2.07%. Among the Hb variants, HbD syndromes were the most common with 99 out of 175 cases (56.57%), making it the most frequently occurring Hb variant in Sindh.

Following HbD, sickle cell disorders were the next most prevalent Hb variant,with a frequency of 56 (32%), followed by HbQ in 12 (6.86%), HbE in 7 (4%), and HbC in 1 case (0.57%). There were significant differences in the distribution of hemoglobinathies by age and gender among cases with β-thalassemia trait, HbD and HbQ syndromes. Figs 1 and 2 illustrate the distribution of patients with Hb variants across various districts in Sindh.

### Clinical presentation and haematological profile of HbD and HbQ syndromes

The mean age at diagnosis for patients with HbD syndrome was 11.6 ± 11 years, with a median age of 8 years, ranging from 4 months to 45 years. For HbQ syndrome patients, the mean age at diagnosis was 20.2 ± 5 years (S1 Table).

**Table 1. Prevalence of Haemoglobinopathies by age and gender in Sindh, Pakistan.**

| Diagnosis | <12 years | | >12 years | | Total N (%) | P value |
|---|---|---|---|---|---|---|
| | Male N (%) | Female N (%) | Male N (%) | Female N (%) | | |
| Normal (AA) | 1069 (22.34) | 670 (14.00) | 390 (8.15) | 1091 (22.81) | 3220 (67.30) | <0.0001 |
| Thalassemia | | | | | | |
| β-thalassemia trait | 128 (2.68) | 81 (1.69) | 176 (3.68) | 297 (6.21) | 682 (14.26) | <0.0001 |
| β- thalassemiamajor | 295 (6.17) | 282 (5.90) | 5 (0.10) | 8 (0.17) | 590 (12.34) | 0.366 |
| Hb Variants | | | | | | |
| HbD syndromes | 45 (0.94) | 13 (0.27) | 13 (0.27) | 28 (0.59) | 99 (2.07) | <0.0001 |
| HbS syndromes | 25 (0.52) | 17 (0.36) | 7 (0.14) | 7 (0.14) | 56 (1.16) | 0.533 |
| HbQ syndromes | 4 (0.08) | 1 (0.02) | 0 (0) | 7 (0.14) | 12 (0.24) | 0.01 |
| HbE syndromes | 0 (0) | 3 (0.06) | 2 (0.04) | 2 (0.04) | 7 (0.14) | 0.429 |
| HbC disease | 0 (0) | 0 (0) | 1 (0.02) | 0 (0) | 1 (0.02) | 1 |
| Hereditary persistence of fetal Hb | | | | | | |
| | 58 (1.21) | 49 (1.02) | 4 (0.08) | 5 (0.10) | 116 (2.41) | 0.732 |
| Total cases of hemoglobinopathies | 555 (11.60) | 446 (9.32) | 208 (4.33) | 354 (7.40) | 1563 (32.70) | <0.0001 |

N: frequency, %: percentage

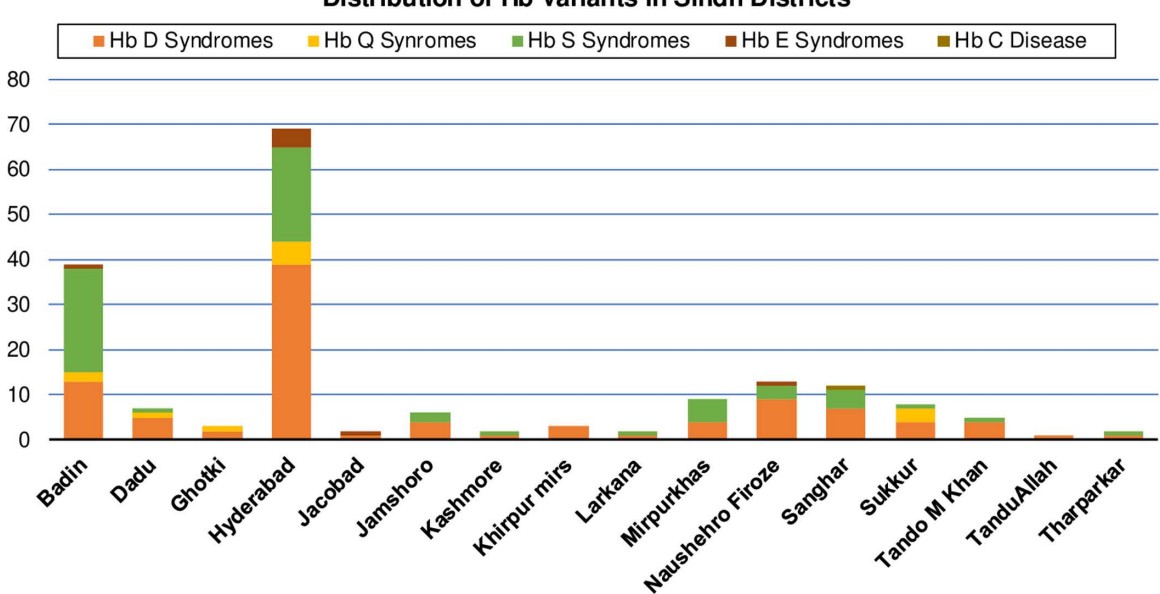

**Fig 2. Bar graph displaying the distribution of hemoglobinopathies across various districts of Sindh, Pakistan.**

Among those with HbD syndrome, most cases were HbD trait (n = 77). Twenty-nine patients presented with severe anemia, 23 had iron deficiency, and 1 had megaloblastic anemia. Approximately 80% showed hypochromic microcytic indices, with 4 patients exhibiting HbF levels exceeding 6.0%. HbD disease was the second most common presentation, involving 12 patients with a mean diagnostic age of 17 ± 8.6 years; 3 of these patients had severe anemia, while 2 exhibited iron deficiency and a hypochromic microcytic blood picture. Five patients were diagnosed with HbDβ thalassemia, all presenting with hypochromic microcytic blood picture and 3 showing severe anemia. HbA2 values varied, and HbF values were elevated, with the highest recorded value being 51.1%. Variations in HbD and HbA levels were observed, likely due to β⁰ and β⁺ mutations, with values ranging between 7.0–39.6% and 5.7–64.4%, respectively. Two patients were identified with β⁰ mutation, indicated by significantly low HbA levels, a history of blood transfusions, and mild splenomegaly.

Four male patients were diagnosed with compound heterozygous HbSD disease, with a mean diagnostic age of 14.7 years. They exhibited marked pallor and anemia, with a mean Hb level of 6.1 ± 1.9 g/dl. Peripheral blood films showed the presence of sickle cells, and half of these patients displayed a hypochromic microcytic blood picture with nucleated red blood cells (NRBCs). HbF levels were moderately elevated in these patients.

Twelve patients undergoing CE-HPLC showed an unknown peak at a common RT of 4.68 minutes and were subsequently confirmed as HbQ India. Of these 9 were HbQ trait cases, and 2 were diagnosed with HbQβ thalassemia. One 19 year old female patient was diagnosed with HbDQ disease, and presented with mild splenomegaly and a history of blood transfusions during childhood. Family studies indicated her father was an HbD trait carrier, and mother was a carrier of HbQ India.

Sixty percent of the patients were diagnosed in adolescence, with 90% presenting with anemia, while iron deficiency was observed in 4 cases. Eight patients with HbQ had anemia, and 2 had a history of blood transfusions during infancy, with improved Hb levels after 6 months of age. Iron deficiency was detected in 4 HbQ trait cases, while patients with HbQβ thalassemia had lower MCV and MCH values. Elevated RBC counts and HbA2 levels in HbQβ thalassemia, differentiated it from HbQ trait. Hematological and CE-HPLC parameters of patients are presented in Tables 2 and 3, respectively.

**Table 2. Haematological parameters in HbD and HbQ syndromes.**

| Parameters | HbD trait | HbD disease | HbDβ thalassemia | HbSD disease | HbDQ disease | HbQ trait | HbQβ thalas-semia |
|---|---|---|---|---|---|---|---|
| Frequency (N = 110; %) | 77 (70) | 12 (10.9) | 5 (4.5) | 4 (3.6) | 1 (0.9) | 9 (8.2) | 2 (1.8) |
| Hb (g/dL) (Mean±SD, Range) | 8.5±3.2 (1.7-16.0) | 10.5±2.7 (5.7-13.8) | 6.8±3.7 (1.6-11.8) | 6.17±1.9 (4.5-8.7) | 12.2 | 8.3±3.2 (3.0-11.9) | 11.9±0.0 |
| HCT (%) (Mean±SD, Range) | 28.1±10.0 (5.7-49.9) | 32.9±8.5 (17.2-44.0) | 21.8±11.6 (5.1-37.2) | 19.4±5.8 (12.7-24.4) | 35.5 | 28.6±9.4 (13.3-42.5) | 40.2±3.1 (38.0-42.5) |
| RBC count ($10^{12}$/L) (Mean±SD, Range) | 4.13±1.5 (0.5-7.1) | 5.39±1.7 (2.4-7.6) | 3.04±1.6 (0.7-5.3) | 2.65±1.0 (1.3-3.7) | 4.55 | 4.12±1.4 (1.5-5.8) | 6.15±0.4 (5.8-6.4) |
| MCV(fl) (Mean±SD, Range) | 70.6±14.5 (49.7-117.1) | 64.0±15.1 (46.2-93.8) | 72.0±2.7 (69.1-75.4) | 76.2±12.7 (65.9-94.1) | 78.0 | 71.3±12.8 (55.6-98.7) | 65.7±9.4 (59.1-72.4) |
| MCH (pg) (Mean±SD, Range) | 21.6±5.8 (12.1-35.0) | 20.7±6.0 (13.8-33.8) | 22.4±1.1 (21.7-24.4) | 24.9±7.7 (18.4-34.8) | 26.8 | 20.2±4.3 (14.4-24.9) | 19.4±1.2 (18.5-20.3) |
| MCHC (g/dL) (Mean±SD, Range) | 30.2±3.5 (22.2-38.4) | 32.1±2.1 (29.3-36.0) | 31.1±1.3 (29.0-32.8) | 32.1±4.9 (27.8-37.0) | 34.4 | 28.8±3.9 (22.6-33.1) | 29.6±2.3 (28.0-31.3) |

N: frequency, %: percentage, SD: Standard deviation

Hb: hemoglobin, HCT: hematocrit, RBC: red blood cell, MCV: mean corpuscular volume, MCH: mean corpuscular hemoglobin, MCHC: mean corpuscular hemoglobin concentration

**Table 3. CE-HPLC parameters in HbD and HbQ syndromes.**

| Parameters | HbD trait | HbD disease | HbDβ thalassemia | HbSD disease | HbDQ disease | HbQ trait | HbQβ thalassemia |
|---|---|---|---|---|---|---|---|
| Frequency (N = 110; %) | 77 (70) | 12 (10.9) | 5 (4.5) | 4 (3.6) | 1 (0.9) | 9 (8.2) | 2 (1.8) |
| HbA (%) (Mean±SD, Range) | 69.9±8.6 (47.6-93.6) | 8.3±3.1 (4.6-13.9) | 46.3±24.0 (5.7-64.4) | 3.07±2.0 (1.2-5.4) | 53.0 | 77.6±6.9 (60.3-85.0) | 82.4±4.0 (79.6-85.3) |
| HbA2 (%) (Mean±SD, Range) | 1.48±0.4 (0.5-2.6) | 2.08±0.8 (1.1-3.5) | 3.4±0.3 (2.8-3.7) | 2.6±1.0 (1.3-3.9) | 1.2 | 2.5±1.4 (1.0-5.4) | 5.4±0.07 (5.4-5.5) |
| HbD (%) (Mean±SD, Range) | 27.0±8.8 (4.3-49.2) | 87.6±3.3 (82.5-93.0) | 21.8±14.8 (7.0-39.6) | 46.5±3.4 (42.2-50.5) | 27.9 | – | – |
| HbF (%) (Mean±SD, Range) | 1.48±2.5 (0.1-18.6) | 1.92±1.9 (0.4-6.1) | 29.4±20.6 (0.3-52.3) | 10.7±6.4 (2.0-17.4) | 0.6 | 0.6±0.5 (0.2-1.3) | 0.5±0.2 (0.3-0.7) |
| HbQ (%) (Mean±SD, Range) | – | – | – | – | 10.2 | 19.1±7.4 (13.4-38.2) | 11.6±4.3 (8.5-14.7) |
| HbS (%) (Mean±SD, Range) | | | | 37.0±3.5 (33.7-42.0) | | | |

N: frequency, %: percentage, SD: Standard deviation

CE-HPLC: Cation Exchange High performance liquid chromatography, Hb: Haemoglobin

## Molecular analysis

Samples from patients with HbD syndromes were analyzed further through sequencing. The HbD Punjab variant was confirmed in exon 3 of the HBB gene, and chromatogram analysis identified a point mutation at the first base of codon 121 (GAA→CAA), resulting in the substitution of glutamic acid with glutamine. Additionally, 4 other mutations in the HBB gene were identified across 5 samples, including βthalassemia, Hb Hinsdale, Hb Renert, and Hb Takasago (Table 4). Fig 3 displays chromatograms of the HbD Punjab and other rare variants. Fig 4 exhibits CE-HPLC patterns of Hb D Punjab with rare Hb variants.

**Table 4. Genetic, haematological and CE-HPLC parameters in HbD cases with rare Hb variants/mutations.**

| Case No. | 01 | 02 | 03 | 04 | 05 |
|---|---|---|---|---|---|
| Variants | c.27dupG | c.27dupG | c.420T>G | c.401T>C | c.397A>G |
| Mutation Type | Insertion | Insertion | Point mutation | Point mutation | Point mutation |
| Exon | Exon 1 | Exon 1 | Exon 3 | Exon 3 | Exon 3 |
| Protein | p.Ser10 Valfs*14 | p.Ser10 Valfs*14 | p.Asn140Lys | p.Val134Ala | p. Lys106>Glu |
| Hb Variant | – | – | Hb Hinsdale | Hb Renert | Hb Takasago |
| β-thalassemia mutation | β- thalassemia mutation | β-thalassemia mutation | – | – | – |
| Age (years) | 28 | 10 | 3 | 3 | 29 |
| Gender | Male | Female | Male | Male | Male |
| Hb (g/dl) | 13.3 | 2.6 | 10.1 | 1.6 | 15.7 |
| MCV(fl) | 58.0 | 61.2 | 62.0 | 59.6 | 69.1 |
| Hb D (%) | 81.7 | 30.1 | 32.5 | 24.4 | 28.0 |
| Hb A (%) | 13.9 | 68.3 | 65.6 | 4.0 | 70.3 |
| Hb A2 (%) | 2.4 | 1.2 | 1.6 | 5.0 | 1.5 |
| Hb F (%) | 2.0 | 0.4 | 0.3 | 66.6 | 0.2 |
| Diagnosis (CE-HPLC) | HbD disease | HbD trait | HbD trait | HbDβ thalassemia | HbD trait |
| Diagnosis (Molecular) | HbDβ-thalassemia | HbDβ- thalas-semia | Compound heterozygous | Compound heterozygous | Compound heterozygous |

Hb: Haemoglobin, MCV: Mean corpuscular volume, CE-HPLC: Cation exchange high performance liquid chromatography

## Discussion

Pakistan, a densely populated South Asian country comprises at least 18 ethnic groups with more than 60 spoken languages leading to significant genetic heterogeneity. Sindh, the second-largest province, holds a diverse genetic landscape shaped by historical events like the Muslim conquest and British rule. The migration during Pakistan's independence further played a pivotal role in shaping Sindhi ethnicity. Notably, Sindhi individuals share genetic traits with Greeks and Georgians, while the Urdu-speaking ethnic community in Sindh displays a heterogeneous Indian ancestry [17,18]. Aditionally, rural Sindh is major endemic region for malaria [19], suggesting a high frequency of haemoglobinopathies [20]. Due to the increased prevalence of consanguineous marriages in interior Sindh [21] there is a further rise in the prevalence of homozygous and compound heterozygous haemoglobinopathy cases [22,23]. These genetic variations significantly contribute to varying risks of developing disorders and influencing disease progression. Ethnic groups in Sindh showcase unique thalassemia and Hb variants [24,25]. Despite worldwide studies on Hb variants [26–29], there is a notable gap in understanding and reporting the prevalence, phenotypic spectrum, hematological and molecular characterization of HbD and Q syndromes and other hemoglobinopathies in Sindh, Pakistan. Furthermore, given the high prevalence of β-thalassemia carriers in Pakistan, the study may uncover novel genotypic interactions between HbD and β-thalassemia, HbS, or other structural variants, contributing to the global understanding of compound hemoglobinopathies.

HbD can exist in many genotypic forms and HbD Punjab is the most common variant (HBB:c.364G>C) [5,30]. Also known as HbD Los Angeles, this variant is predominantly found in regions such as Punjab (India), Italy, Austria, Belgium, and Turkey [10,11]. There are six other HbD variants identified as HbD Iran: beta 22(B4) Glu>Gln; HBB:c.67G>C, HbD Agri: beta 9(A6) Ser>Tyr and beta 121(GH4) Glu>Gln; HBB:c.(29C>A;364G>C), HbD Ibadan: beta 87(F3) Thr>Lys; HBB:c.263C>A, HbD Bushman: beta 16(A13) Gly>Arg; HBB:c.49G>C, HbD Neath: beta 121(GH4) Glu>Ala;

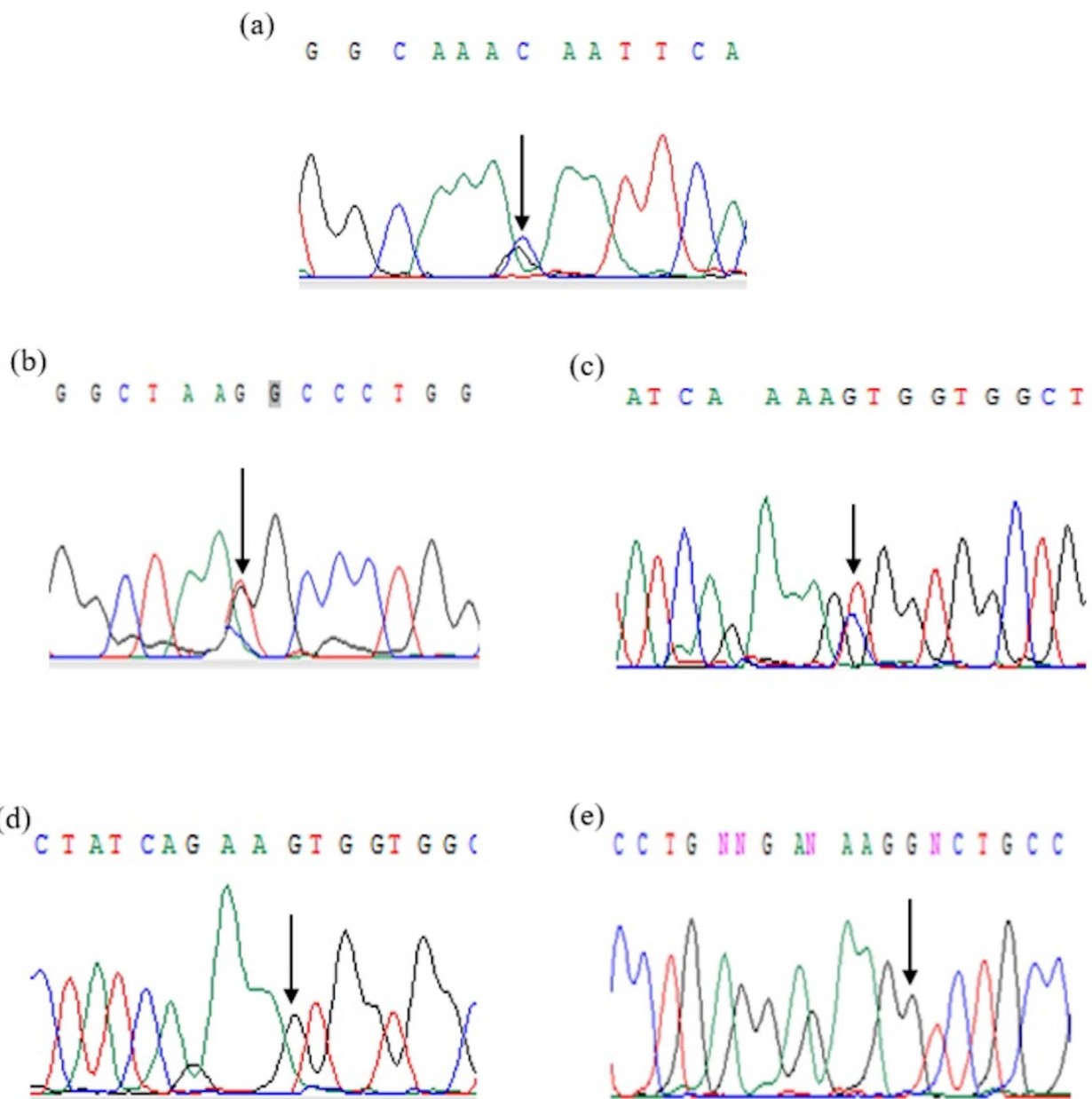

**Fig 3. Chromatograms of HBB gene variants showing. (a)** c.364G>C, p.Glu122Gln **(b)** c.420T>G, p.Asn140Lys **(c)** c.401T>C, p.Val134Ala **(d)** c.397A>G, p.Lys106* **(e)** c.27dupG, p.Ser10 Valfs*14.

HBB:c.365A>C, HbD Ouled Rabah: beta 19(B1) Asn>Lys; HBB:c.(60C>A or 60C>G) and HbD Granada: beta 22(B4) Glu>Val;HBB:c.68A>T [5]. HbD Punjab and HbD Iran have same electrophoretic mobility but can be differentiated on CE-HPLC and molecular analysis [31].

In this study, HbD was the most prevalent Hb variant in Sindh, followed by HbS and HbQ, with HbE and HbC being less common. HbD constituted 56.57% of all Hb variants, with most cases being the HbD Punjab variant. This aligns with findings from Xinjiang, China, and Turkey, where HbD is also the most common variant, with frequencies of 55.6% and 57.9%, respectively. In India, HbD Punjab has a frequency of about 2% [32] reaching approximately 3% among Sikhs in

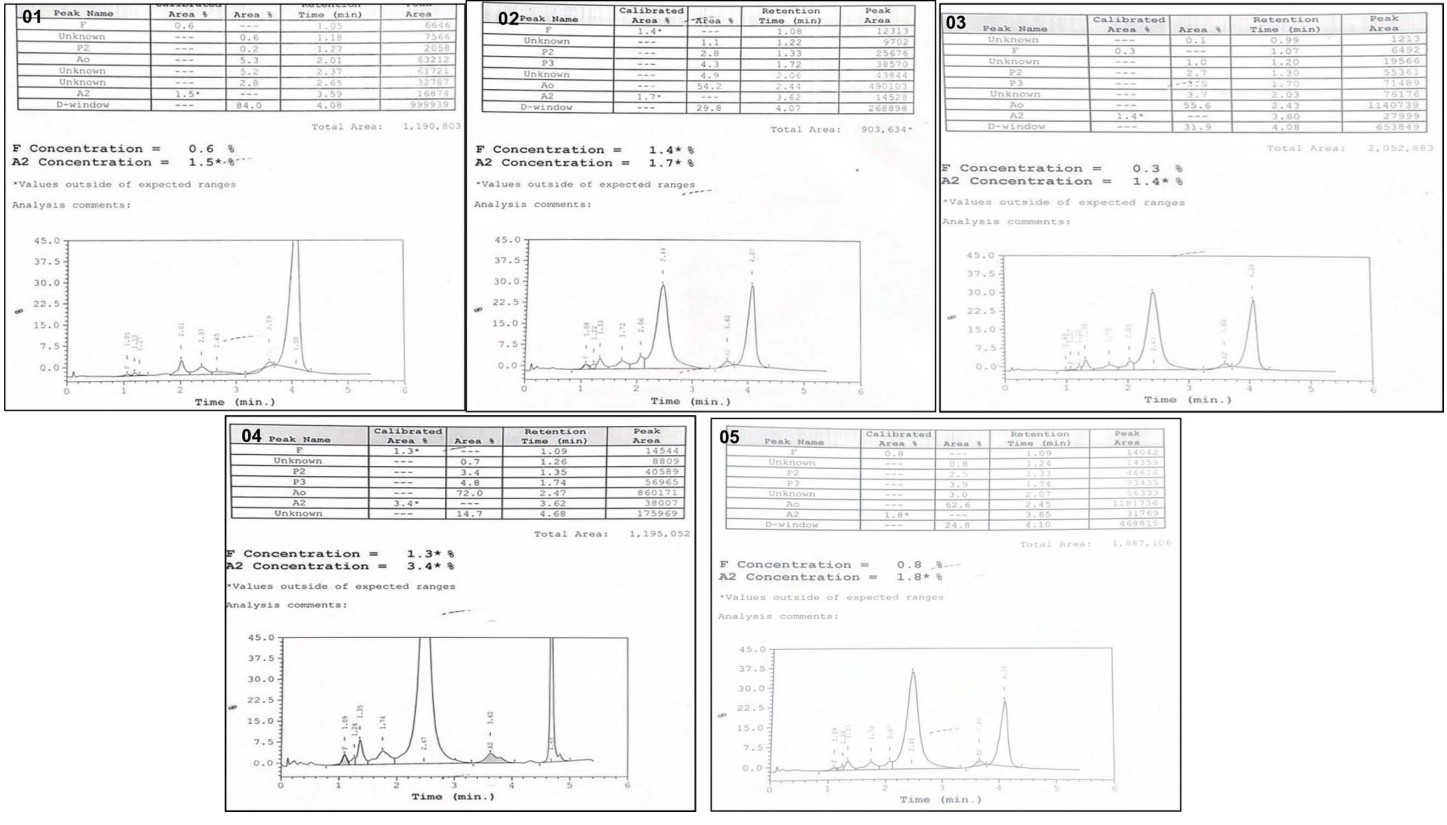

**Fig 4. CE-HPLC patterns of Hb D Punjab with rare Hb variants exhibiting case no 1, 2, 3, 4 & 5 (** Table 4**).**

northwestern regions and 55% among Hb variants [33], similar to our results. The frequency of Hb variants across other provinces in Pakistan showed that HbD Punjab is most common in Punjab, with a frequency of 69.6%, while HbS is the most prevalent in Khyber Pakhtunkhwa and Balochistan, with frequencies of 85.2% and 89.1%, respectively [8].

Both homozygous and heterozygous HbD Punjab patients are typically asymptomatic, though some may exhibit microcytic hypochromic anemia. When combined with other Hb variants, moderate to severe disease can occur [32]. For instance, homozygous HbS or its combination with β-thalassemia, HbD, or HbC often leads to severe clinical presentations. In HbD disease, a high proportion of red blood cells (RBC) containing HbD can reduce RBC size and count, contributing to mild anemia. While HbDD, the rarest form, is mostly asymptomatic, it may occasionally present with mild or moderate hemolytic anemia. HbD trait carriers do not develop HbD disease or sickle cell but can pass these traits to future generations, resulting in HbD, HbSD, or HbD/β-thalassemia disease. When one parent has the HbD trait and the other carries the β/0 thalassemia or sickle cell trait, there is a 25% chance of HbD/β0 thalassemia (Dβ0) or sickle cell disease in each pregnancy. These conditions are a lifelong ailment associated with significant health challenges. Although molecular studies are consistently recommended as conclusive investigations, they are not frequently pursued in the majority of cases [34]. In this study, 77 patients were diagnosed with HbD trait, 12 with HbD disease (DD), 5 with HbDβ thalassemia, 4 with HbSD, and 1 with HbDQ disease. Shanthala Devi AM, et al [35] reported 12 cases (2.0%) of HbD, 6 HbSD, 1 HbDD, 1 HbDβ thalassemia and 4 HbD trait among 589 cases of haemoglobinopathies. Most patients in our study presented between 6 months and 12 years, often with hypochromic microcytic indices, including 26 cases with iron deficiency. The high prevalence of iron deficiency in Pakistan [36], contributes to delayed HbD diagnosis, often until severe anemia occurs [37].

HbDβ-thalassemia cases show severe anemia, reduced erythrocytic indices, and a hypochromic, microcytic red cell picture. In contrast, HbSD cases often present with pallor, frequent pain crises, and splenomegaly [38]. In our study, all affected individuals experienced severe anemia and RBC sickling. The notable prevalence of HbD along with HbS, and their moderate to severe clinical manifestations underscores the importance of screening, genetic counseling, and prenatal diagnosis in our population.

Additionally, four other notable mutations were found that were not identified on CE-HPLC. The mutation HBB: c.27dupG, p.Ser10 Valfs*14 was identified in two cases, one patient exhibited severe hypochromic microcytic anemia and required transfusions. This mutation, associated with transfusion dependent β-thalassemia major and sickle cell anemia [39], explains the severe anemia in a patient with the HbD trait. Hb Renert, a rare hemoglobin variant with the point mutation c.401T>C, p.Val134Ala, was identified in a patient diagnosed with HbDβ thalassemia on CE-HPLC. The patient presented with severe anemia and elevated HbF and HbA2 levels. Hb Renert is a neutral variant linked to chronic hemolysis and is detectable by reverse-phase HPLC [40]. Hb Hinsdale, another rare variant, was identified in a patient diagnosed with HbD trait, showing only mild anemia [40], consistent with findings in our study. Though it did not show a characteristic band on CE-HPLC, it was detected during molecular analysis, consistent with previous studies [41]. Hb Takasago, identified in a patient with HbD trait, showed a point mutation c.397A>G, p.Lys106*. Originally identified in a Japanese female, this variant is clinically asymptomatic [42]. The prevalence of HbD along with other Hb variants combined with moderate to severe clinical manifestations, highlight the necessity of screening, molecular diagnosis, and prenatal diagnosis in the Sindhi, Pakistani population.

HbQ is a rare α1 globin gene variant with three molecular variants. HbQ India: α1 64(E13) Asp>His; HBA1:c.193G>C, HbQ Iran: α2 or α1 75(EF4) Asp>His; HBA2:c.226G>C (or HBA1), and HbQ Thailand: α1 74(EF3) Asp>His; HBA1:c.223G>C (5). The prevalence of HbQ India has been reported between 0.2% and 0.4% in India and Nepal [4]. In this study, all 12 patients were identified with HbQ India: 9 with HbQ trait, 2 as compound heterozygous for HbQβ thalassemia, and 1 female as compound heterozygous for HbDQ disease. Iron deficiency was observed in 4 HbQ trait cases though MCV and MCH values were lower in HbQβ thalassaemia. Elevated RBC count and HbA2 levels in HbQβ thalassaemia helped distinguish it from HbQ trait. Krishna et al. [43] noted that the α2 gene becomes dominant between 24–36 weeks of gestation, with the α2/α1 ratio increasing and γ globin synthesis rapidly declining, supporting our observation of Hb improvement post-infancy. Phanasgaonkar et al. [44] documented 36 HbQ India trait cases, 22 with HbQ India and β-thalassemia trait, 3 with HbQ India β-thalassemia major, and 3 homozygous HbQ India cases in a study of 64 patients. Although typically asymptomatic, HbQ India can present clinical manifestations in compound heterozygous and homozygous states with other hemoglobinopathies.

## Conclusion

In conclusion, HbD Punjab stands out as the predominant Hb variant in the Sindhi Pakistani population, followed by HbS and HbQ. Clinical outcomes are influenced by the co-occurrence of thalassemia, HbS, rare Hb variants and nutritional deficiencies. While CE-HPLC is valuable for diagnosis; molecular analysis is crucial for confirming HbD and detecting rare mutations in patients with atypical features. Detection of HbD through screening programs, along with antenatal diagnosis, genetic counseling, and family studies, can significantly aid in the prevention of sickling disorders and other hemoglobinopathies.

## Supporting information

**S1 Table. Clinical Parameters in HbD and HbQ syndromes.** N: frequency, %: percentage, SD: Standard deviation. (DOCX)

**S1 Abstract Graphic. Tif**
(TIF)

## Author contributions

**Conceptualization:** Sunila Tashfeen, Ikram Din Ujjan.

**Data curation:** Sunila Tashfeen, Ikram Din Ujjan.

**Formal analysis:** Sunila Tashfeen, Hina Shaikh, Muhammad Arif Sadiq, Feriha Fatima Khidri, Ali Raza Rajput.

**Investigation:** Sunila Tashfeen, Hina Shaikh, Ali Raza Rajput.

**Methodology:** Sunila Tashfeen, Ikram Din Ujjan, Muhammad Arif Sadiq.

**Project administration:** Sunila Tashfeen, Hina Shaikh.

**Resources:** Sunila Tashfeen.

**Supervision:** Ikram Din Ujjan, Ali Muhammad Waryah.

**Validation:** Sunila Tashfeen, Ali Muhammad Waryah.

**Visualization:** Sunila Tashfeen, Feriha Fatima Khidri.

**Writing – original draft:** Sunila Tashfeen.

**Writing – review & editing:** Ikram Din Ujjan, Hina Shaikh, Muhammad Arif Sadiq, Feriha Fatima Khidri, Ali Raza Rajput, Ali Muhammad Waryah.

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
