## [Decision Letter · Decision Letter 0]

9 Oct 2024

PONE-D-24-40816Clinicohematological and Molecular Analysis of Hemoglobin D Syndrome and Unknown Variants in the Hemoglobinopathy Spectrum of Sindh, PakistanPLOS ONE

Dear Dr. Tashfeen,

Thank you for submitting your manuscript to PLOS ONE. After careful consideration, we feel that it has merit but does not fully meet PLOS ONE’s publication criteria as it currently stands. Therefore, we invite you to submit a revised version of the manuscript that addresses the points raised during the review process.

**ACADEMIC EDITOR: ** I have completed my evaluation of your manuscript as well as the reviewers have reviewed the manuscript. You are advised to incorporate the information as suggested by the reviewers and resubmit the manuscript. It requires major revision. Please submit your revised manuscript by Nov 23 2024 11:59PM. If you will need more time than this to complete your revisions, please reply to this message or contact the journal office at plosone@plos.org . Please include the following items when submitting your revised manuscript:

We look forward to receiving your revised manuscript.

Kind regards,

Kamlesh Kumari

Academic Editor

PLOS ONE

3. In the online submission form you indicate that your data is not available for proprietary reasons and have provided a contact point for accessing this data. Please note that your current contact point is a co-author on this manuscript. According to our Data Policy, the contact point must not be an author on the manuscript and must be an institutional contact, ideally not an individual. Please revise your data statement to a non-author institutional point of contact, such as a data access or ethics committee, and send this to us via return email. Please also include contact information for the third party organization, and please include the full citation of where the data can be found.

4. We note that Figure 2 in your submission contain [map/satellite] images which may be copyrighted. All PLOS content is published under the Creative Commons Attribution License (CC BY 4.0), which means that the manuscript, images, and Supporting Information files will be freely available online, and any third party is permitted to access, download, copy, distribute, and use these materials in any way, even commercially, with proper attribution. For these reasons, we cannot publish previously copyrighted maps or satellite images created using proprietary data, such as Google software (Google Maps, Street View, and Earth). For more information, see our copyright guidelines: http://journals.plos.org/plosone/s/licenses-and-copyright.

1. You may seek permission from the original copyright holder of Figure 2 to publish the content specifically under the CC BY 4.0 license. 

Additional Editor Comments:

I have completed my evaluation of your manuscript as well as the reviewers have reviewed the manuscript. You are advised to incorporate the information as suggested by the reviewers and resubmit the manuscript. It requires major revision.

Reviewers' comments:

Reviewer's Responses to Questions

**Comments to the Author**

1. Is the manuscript technically sound, and do the data support the conclusions?

Reviewer #1: Yes

Reviewer #2: Partly

2. Has the statistical analysis been performed appropriately and rigorously? 

Reviewer #1: No

Reviewer #2: N/A

3. Have the authors made all data underlying the findings in their manuscript fully available?

Reviewer #1: Yes

Reviewer #2: No

4. Is the manuscript presented in an intelligible fashion and written in standard English?

Reviewer #1: Yes

Reviewer #2: No

5. Review Comments to the Author

Reviewer #1: The study investigated the prevalence and molecular characteristics of hemoglobin D (HbD) syndrome and other hemoglobin variants in Sindh, Pakistan. It identifies HbD Punjab as the most common variant, followed by sickle cell disorders and HbQ. The research highlights the importance of molecular analysis for accurate diagnosis and suggests integrating HbD detection into screening programs to prevent hemoglobinopathies. The findings emphasize the need for genetic counseling and prenatal diagnosis to manage and reduce the burden of these disorders. The manuscript could benefit from some minor revision points:

1. Authors could provide a graphical abstract to summarizes the work graphically, potentially stimulating researchers to view the published manuscript.

2. There are some areas where sentence structure could be improved for clarity and flow. Consider a thorough revision for grammatical consistency also.

3. More contextual background could be added to justify why this region is a focal point of the study. The mention of Sindh's genetic diversity is insightful, but adding data on the prevalence of hemoglobinopathies in the region (and comparison to other regions in Pakistan) would strengthen the rationale for the study.

4. The images uploaded are blurred and are of low quality, authors should improve the quality of the images.

5. In Table 1, 2, 3 and 4 clearly describe the values in brackets “( )” in caption or below the table.

6. The methodology is well-described, but some subsections could use additional detail. For instance, the inclusion and exclusion criteria could be expanded, especially regarding patients with prior blood transfusions (time period of prior transfusion), as this might significantly affect results. Additionally, clarify whether there was any control group or comparative analysis with other populations.

7. Consider providing more statistical depth, especially in comparing various hemoglobinopathies and their prevalence among age and gender groups. This would lend additional robustness to the data and findings.

8. It will be beneficial to include more recent studies, particularly those published within the last 4-5 years, to reflect the current state of research on hemoglobinopathies.

Reviewer #2: The manuscript describes a series of variants in the Pakistani population. There are following observations on the manuscript:

Methods section:

1. Complete the HPLC method as CE-HPLC

2. Add the manufacturer details for the Capillary sequencing platform. Replace Sangers sequencing by Capillary sequencer.

3. Primers for the ARMS PCR should be added as well and the gel picture can be added as a Supplementary figure.

Results section:

1. The authors have described 3 Hb variants and 1 beta thal mutation in 2 patients. The diagnosis given in table 5 do not tally with the actual diagnosis. ex. Patient 1 has Dbeta thalassemia and not HbD disease, patient 2 also has HbDbeta thalassemia and not HbD Trait. Patient 3 is compound heterozygous for HbD and HbHinsdale. The patient 4 is a case of compound heterozygous for HbD and HbRenert and not Dbeta thalassemia. Patient 5 also has compound heterozygous for HbD and HbTakasago and not HbDTrait.

2. The above may also be a reason for the severity of anemia observed for patient 4 as HbRenert is highly unstable. Was a test for unstable hemoglobins performed for this patient?

3. The results are unclear about the data on HbD syndromes. Are the authors describing the cases after completion of molecular analysis with the genetic classification or are describing cases after the initial HPLC results.

4. Severe anemia in many patients remains unexplained in all the conditions.

5. Cases of compound heterozygous HbD/Q-India have shown mild anemia and not required any transfusions. Hence, the transfusion in the case described needs to be investigated. The same is true of splenomegaly in this patient.

6. Table 2 can be submitted as a supplementary table.

7. The way of writing Hb variants should be as per the HGVS nomenclature throughout the manuscript. HBB:c.420T>G, p.Asn140Lys and so on. The authors seem to switch from one format to the other.

8. rs35699606 should also be expressed in HGVS nomenclature. It is the Cd8/9 mutation. It should be expressed as c.27dupG in preference to the nomenclature used.

9. The authors should present the chromatograms of these cases with the Sanger sequencing data. Was the case 4 submitted for any additional analysis such as Capillary electrophoresis? This is an intriguing case.

Discussion:

1. Please use HGVS nomenclature for all the variants.

2. It requires significant shortening.

6. PLOS authors have the option to publish the peer review history of their article (what does this mean? ). If published, this will include your full peer review and any attached files.

**Do you want your identity to be public for this peer review?** For information about this choice, including consent withdrawal, please see our Privacy Policy .

Reviewer #1: No

Reviewer #2: **Yes: ** Jasmita

---

## [Author Response · Author response to Decision Letter 1]

11 Nov 2024

Dear Reviewers,

Thank you so much for the valuable feedback and insightful comments. I have addressed these issues accentuated by respected reviewers and amended in manuscript as advised. These are as follows:

Reviewer #1:

1. Graphical abstract with revised manuscript is uploaded.

2. We have thoroughly revised manuscript for language usage, spelling, and grammar in revised draft

3. We have added a comparison of hemoglobin variant frequencies across provinces in Pakistan in the discussion section (Ref: Israr AS, et al.), to provide additional contextual background.

4.We have enhanced the image quality using Apex CoVantage PACE 3.0.2.0, adjusted the image dimensions, and converted the JPG/PDF files to high-resolution TIF files for improved clarity.

5.We have clearly described the values below Tables 1, 2, 3, and 4 as requested.

6.We have further elaborated on the exclusion criteria, specifying that patients with blood transfusions within the last 3 months, known cases of thalassemia, and those undergoing repeat testing were excluded from the study. While no control group was included, we conducted comparative analyses with national and international studies, incorporating local, provincial, and regional statistics.

7.We have conducted statistical analyses on the distribution of hemoglobinopathies by age and gender using the chi-square test/Fisher’s exact test, as applicable (Table 1) in the revised manuscript.

8.We have included recent studies: References: 2, 3, 6-9, 17-32, 35-37, 40, to reflect current research on hemoglobinopathies in our manuscript.

Reviewer #2: Method Section

1.HPLC is replaced by CE-HPLC method in manuscript.

2. Manufacturer details for the Capillary sequencing platform is added and Sangers sequencing is replaced by Capillary sequencer.

3.The primers for the ARMS PCR used in this study were based on the methods described by Abraham R. et al. The reference has been added in the revised manuscript.

Result Section

1.We have revised the table 4 (previously table 5) to accurately reflect each patient's diagnosis based on both hematological findings and molecular results, as suggested, in the revised manuscript.

2.We did not perform a specific test for unstable hemoglobins for this patient. We recognize that such testing could provide additional insights into the severity of anemia, and we will consider this in future analyses.

3.In the results section, we are describing the cases based on the findings after the completion of both the initial CE-HPLC results and subsequent molecular analysis. We aimed to provide a comprehensive overview that includes both the hematological data and genetic confirmation to ensure accurate diagnosis.

4.Severe anemia was observed in 41 cases, which can largely be explained by the presence of a compound heterozygous state and nutritional deficiencies in these patients. Specifically, 30 patients had iron deficiency anemia, and 1 patient had B12 and folate deficiency.

5.While most cases of compound heterozygous HbD/Q-India typically exhibit mild anemia without the need for transfusions, our study identified a case where transfusions were necessary during childhood. The variability in clinical presentation may be attributed to several factors, including the presence of coexisting conditions, such as nutritional deficiencies or infections, that can exacerbate anemia. Our findings suggest that while mild anemia is common, unique circumstances in certain patients can lead to more severe manifestations requiring transfusions, which should be explored on a case-by-case basis.

6.Table 2 is added as a supplementary table in the revised manuscript.

7 & 8.The variants have been updated as per the HGVS nomenclature in the revised manuscript.

9.Chromatograms and HPLC results for the rare Hb variants have been added to the article. Case 4 was not submitted for capillary electrophoresis, as this analysis was not part of our methodology. The final diagnosis for this case was confirmed through molecular analysis. While CE was not part of our methodology, we appreciate the potential benefits it could bring and will consider it for future studies.

Discussion

1.The variants have been updated as per the HGVS nomenclature in the revised manuscript.

2.The discussion section has been shortened in the revised manuscript.

---

## [Decision Letter · Decision Letter 1]

26 Dec 2024

PONE-D-24-40816R1Clinicohematological and molecular analysis of hemoglobin D syndrome and unknown variants in the hemoglobinopathy spectrum of Sindh, PakistanPLOS ONE

Dear Dr. Tashfeen,

Thank you for submitting your manuscript to PLOS ONE. After careful consideration, we feel that it has merit but does not fully meet PLOS ONE’s publication criteria as it currently stands. Therefore, we invite you to submit a revised version of the manuscript that addresses the points raised during the review process.

**ACADEMIC EDITOR: The reviewers have recommend for the reconsideration of the manuscript.**

We look forward to receiving your revised manuscript.

Kind regards,

Kamlesh Kumari

Academic Editor

PLOS ONE

Journal Requirements:

Additional Editor Comments:

Reviewers have submitted their reports on manuscript and recommend for the revision.

Reviewers' comments:

Reviewer's Responses to Questions

**Comments to the Author**

1. If the authors have adequately addressed your comments raised in a previous round of review and you feel that this manuscript is now acceptable for publication, you may indicate that here to bypass the “Comments to the Author” section, enter your conflict of interest statement in the “Confidential to Editor” section, and submit your "Accept" recommendation.

Reviewer #1: All comments have been addressed

Reviewer #3: (No Response)

2. Is the manuscript technically sound, and do the data support the conclusions?

Reviewer #1: Yes

Reviewer #3: Partly

3. Has the statistical analysis been performed appropriately and rigorously? 

Reviewer #1: N/A

Reviewer #3: Yes

4. Have the authors made all data underlying the findings in their manuscript fully available?

Reviewer #1: Yes

Reviewer #3: Yes

5. Is the manuscript presented in an intelligible fashion and written in standard English?

Reviewer #1: Yes

Reviewer #3: Yes

6. Review Comments to the Author

Reviewer #1: (No Response)

Reviewer #3: 1. There is the need to improve the presentation of the figures.

2. The figures are in bad resolution

3. All the methodology subsections should be adequately backed up with a reference

4. A strong problem statement should be provided in the introduction

5. The novelty of the work is missing

7. PLOS authors have the option to publish the peer review history of their article (what does this mean? ). If published, this will include your full peer review and any attached files.

**Do you want your identity to be public for this peer review?** For information about this choice, including consent withdrawal, please see our Privacy Policy .

Reviewer #1: No

Reviewer #3: No

---

## [Author Response · Author response to Decision Letter 2]

23 Jan 2025

Reviewer #3:

Comment 1: There is the need to improve the presentation of the figures.

Response: We have worked and improved the presentation of figures using Apex CoVantage PACE 3.0.2.0.

Comment 2: The figures are in bad resolution

Response: Thank you for your observation. We have improved resolution of all figures according to PLOS guidelines. Resolution of all figures are now between 300-600 dpi and file size is less than 10 MB.

Comment 3: All the methodology subsections should be adequately backed up with a reference

Response: Following references have been added in the methodology in the revised manuscript.

12. Chakma S, Das S, Chakma K. Detection of abnormal haemoglobin variants and its characterization among anaemicsby high performance liquid chromatography (HPLC): A prospective study from North India. European Journal of Molecular and Clinical Medicine. 2022 Jan 30;9(3):2552-62.

13. World Health Organization. Guideline on haemoglobin cutoffs to define anaemia in individuals and populations. World Health Organization; 2024 Mar 5.

15. Newton CR, Graham A, Heptinstall LE. Analysis of any point mutation in DNA. The amplification refractory mutation system (ARMS). Nucl Acids Res 1989;17:2503–16.

Comment 4: A strong problem statement should be provided in the introduction

Response: The problem statement has been added in the introduction of the revised manuscript.

Comment 5: The novelty of the work is missing

Response: Despite extensive global studies on hemoglobin variants, there remains a significant gap in understanding the prevalence, phenotypic spectrum, and molecular characterization of HbD, HbQ syndromes, and other hemoglobinopathies in Sindh, Pakistan. While β-thalassemia has been extensively studied due to its high carrier prevalence in the region, limited data exist on HbD syndromes and their potential interactions with other hemoglobin variants, such as HbS and HbQ. The findings of the study may contribute to the regional and global understanding of hemoglobinopathies, particularly in identifying unique mutation patterns and genotype-phenotype correlations.

The novelty of this work lies in its comprehensive approach to exploring underreported hemoglobinopathies, its focus on a geographically and genetically diverse population of Sindh, Pakistan and its potential to inform diagnostic accuracy, tailored therapeutic strategies, and public health interventions aimed at reducing the burden of hemoglobinopathies in low-resource settings.

The novelty of the study is mentioned in introduction and discussion section of revised manuscript.

---

## [Decision Letter · Decision Letter 2]

18 Feb 2025

Clinicohematological and molecular analysis of hemoglobin D syndrome and unknown variants in the hemoglobinopathy spectrum of Sindh, Pakistan

PONE-D-24-40816R2

Dear Dr. Tashfeen,

We’re pleased to inform you that your manuscript has been judged scientifically suitable for publication and will be formally accepted for publication once it meets all outstanding technical requirements.

Kind regards,

Kamlesh Kumari

Academic Editor

PLOS ONE

Additional Editor Comments (optional):

Accept

Reviewers' comments:

Reviewer's Responses to Questions

**Comments to the Author**

1. If the authors have adequately addressed your comments raised in a previous round of review and you feel that this manuscript is now acceptable for publication, you may indicate that here to bypass the “Comments to the Author” section, enter your conflict of interest statement in the “Confidential to Editor” section, and submit your "Accept" recommendation.

Reviewer #4: (No Response)

2. Is the manuscript technically sound, and do the data support the conclusions?

Reviewer #4: (No Response)

3. Has the statistical analysis been performed appropriately and rigorously? 

Reviewer #4: (No Response)

4. Have the authors made all data underlying the findings in their manuscript fully available?

Reviewer #4: (No Response)

5. Is the manuscript presented in an intelligible fashion and written in standard English?

Reviewer #4: (No Response)

6. Review Comments to the Author

Reviewer #4: I have reviewed the manuscript titled "Clinicohematological and molecular analysis of hemoglobin D syndrome and unknown variants in the hemoglobinopathy spectrum of Sindh, Pakistan." I appreciate the authors' efforts in conducting this valuable research.

After careful consideration, I have no additional comments or concerns regarding the manuscript, including aspects related to dual publication, research ethics, or publication ethics.

7. PLOS authors have the option to publish the peer review history of their article (what does this mean? ). If published, this will include your full peer review and any attached files.

**Do you want your identity to be public for this peer review?** For information about this choice, including consent withdrawal, please see our Privacy Policy .

Reviewer #4: **Yes: ** Asaad Babker

---

## [Editor Report · Acceptance letter]

PONE-D-24-40816R2

PLOS ONE

Dear Dr. Tashfeen,

I'm pleased to inform you that your manuscript has been deemed suitable for publication in PLOS ONE. Congratulations! Your manuscript is now being handed over to our production team.

Kind regards,

on behalf of

Dr. Kamlesh Kumari

Academic Editor

PLOS ONE